# Real-Time Cone-Growth Model for Determination of Pharmaceutical Powder Flow Properties

**DOI:** 10.3390/pharmaceutics16030405

**Published:** 2024-03-15

**Authors:** Gyula Farkas, Sándor Nagy, Attila Dévay, Aleksandar Széchenyi, Szilárd Pál

**Affiliations:** Institute of Pharmaceutical Technology and Biopharmacy, University of Pécs, Rókus Str. 2, H-7624 Pécs, Hungary; farkas.gyula92@gmail.com (G.F.); szechenyi.aleksandar@gytk.pte.hu (A.S.)

**Keywords:** flow properties, static angle of repose, flow time, flow curve, conical section

## Abstract

The flow properties of pellets or granules are crucial for further processing drug dosage forms. Optimal compression or filling of multiparticulate dosage forms into capsules is influenced by forces between discrete particles, which could be partially characterized by flow properties. Several techniques have been developed to examine flowability, including static and dynamic methods applying empirical studies and up-to-date chaos theory; however, the newest methods seem only to be powerful with the supplementation of empirical principles. Our experiments try to refine both the technique of analysis and the methods, by finding new, alternative ways. Our approach to the flowability measurements was to set up a dynamic time-dependent model that combined empirical observations and chaos theory on a geometrical basis, thus finding new characteristics regarding the flow properties of pellets and granules that could be relevant for drug developers. Our findings indicate that sphericity and particle size are the most significant factors influencing the flowability of pharmaceutical multiparticular preparations. Furthermore, this study confirms that integrating chaos theory and empirical observations in a time-dependent dynamic model provides a comprehensive understanding of particle flow behavior, pivotal for optimizing manufacturing processes.

## 1. Introduction

Granules and pellets play important roles as standalone dosage forms, and dosage forms for further processing. Ensuring optimal pharmaceutical-physical properties is necessary in developing a dosage form. The flow properties of multiparticulates are cardinal, due to the need for accurate dosing or further processing. Insufficient flow characteristics of pellets and granules may cause difficulties in additional procedures.

Developments in the measurement of powder flow characteristics need to consider how powder-filling, tableting, or capsule-filling machines operate during the production of pharmaceuticals. In this way, whether the flowability of powders will be adequate during drug production can be well modeled, which is also in line with the principles of QbD.

Flow properties are determined by several factors derived from the nature of particles, such as shape; size; size distribution; surface; purity; crystallinity; interparticle forces (friction forces, surface tension, molecular forces [1]), and external factors such as the force of gravity; temperature; moisture content; electrostatic charge; aeration; and particle-wall interaction [2,3,4,5]. Examining the effect of these factors on powder flowability is still a favored topic, since development in measurement techniques and powder flow theory is on-going. Currently, there is no simple method to determine the complex properties of powder flow for Process Analytical Technology (PAT), which is why developing methods that can be implemented in a PAT system is highly important.

Flow properties of particles could be characterized in a number of ways, applying basic methods like examining the static angle of repose; flow through an orifice; compressibility measurements; shearing tests; observing avalanche behavior; and applying powder rheometers [6,7,8].

Several empirical assessments were made regarding the effect of the properties of particles on powder flow using these methods.

Particles below 50 μm have irregular flow or practically no flow due to van der Waals forces. An increase in particle size usually increases flowability. The flow rate achieves maximum at an orifice diameter-particle diameter ratio of 20–30, and stops completely below 6 and above the size of 1200 μm [9].

Moisture content can affect flow properties too, due to forces usually induced by surface tension. Depending on material properties, low moisture content can lead to the development of electrostatic charges and influence the flow [10].

Several numeric models were created describing this flow through an orifice. These models are often based on a Brown-Richards-type relationship modeling the static flow as follows:(1)$$Q=KA\sqrt{2gh\;\rho t}$$

*Q* is the flow rate, ρt is the powder density, *A* is the cross-sectional area of the funnel outlet, *h* is the height of the powder bed above the outlet, *K* is the material-dependent constant, and *g* is the force of gravity [11]. 

The application of novel techniques on the shape characterization of particles revealed a correlation between flow properties and surface morphology. Benoît Mandelbrot’s theory [12] of fractal geometry led to the first alternative explanation of the influence of shape factors on flow properties [13,14]. Irregularity of shape usually causes poor flowability characteristics. Spherical and smooth particles with an appropriate size have sufficient flow [15]. 

The complexity of the theory of powder flow derives from a novel approach regarding particles behaving as self-organized structures affecting each other. This principle dates back to 1987 when Per Bak, Chao Tang, and Kurt Wiesenfeld created their top-cited publication on self-organized criticality [16]. This theory perfectly fits Mandelbrot’s fractals, which could describe several natural phenomena. Bak et al. took the example of a sandpile with an angle of repose θ. Adding a small amount of sand results in a slight response, but by continuing the sand addition, the pile reaches a state when the local slope of the sand particles is higher than θ, and an avalanche effect takes place, thus aiming for a stable state.

An important step in powder flow theory was the combination of fractal geometry with self-organized criticality, which resulted in the rotating drum technique [17]. This was a relevant milestone in the dynamic measurement and examination of avalanche behavior in detail. This work aims to integrate the angle of repose measurement method with chaos theory principles; our study explores the flowability of powders through the lens of self-organized criticality, offering new insights into the dynamics that govern particulate systems

## 2. Materials and Methods

Pellets and granules of different sizes were produced containing 10% ibuprofen (Hungaropharma, Budapest, Hungary, average particle size 70 μm, residual moisture content 4.2%); 37% α-lactose-monohydrate (DC, BDI, Zwolle, The Netherlands, average particle size 70 μm, residual moisture content 5.1%); 50% microcrystalline cellulose (Avicel PH 101, FMC, Philadelphia, PA, USA, average particle size 50 μm, residual moisture content 3.49%); and 3% ethylcellulose (Hercules, Wilmington, NC, USA, 7 cP viscosity grade with 48.0–49.5 *w*/*w* of ethoxy group). Purified water was used as granulation liquid. The quality of all materials used in the experiments was Ph. Eur.

The categorical factor for pellets and granules was distinguished based on sphericity. Particles with a sphericity above 0.91 were considered pellets, and those with a sphericity equal to or below 0.90 were considered granules.

### 2.1. Production of Samples

Samples were produced according to a central composite experimental design with three numeric (particle size, moisture content, glidant/lubricant content) and one categoric factor (granule/pellet) (Table 1). The categorical factor for pellets and granules was distinguished based on sphericity. Particles with a sphericity above 0.91 were considered pellets, and those with a sphericity equal to or below 0.90 were considered granules. We have set up our experimental design by grounding our parameter selection based on robust scientific literature that identifies particle size, shape, moisture, and inter-particle forces, and we have added glidant excipients as critical determinants of powder flowability. Our methodology integrates a central composite experimental design to explore these parameters’ impact on flow properties systematically. This approach allows us to delineate each variable’s influence and interactions, as supported by significant empirical evidence [18,19,20].

Pellets were produced using a 1000 mL laboratory high-shear mixer (Pro-C-epT 4M8 Granulator, Zelzate, Belgium) with a three-blade impeller and a chopper. Granules were produced using an oscillatory granulator (Erweka GmbH, Heusenstamm, Germany). Samples of both productions were air dried at 40 °C for 1 h. Granules then again passed through the oscillatory granulator according to the desired particle size. Purified water was used as granulation liquid in both cases. Each batch had a total dry mass of 120 g. 

After preparation, samples were stored in closed containers for 24 h at 25 °C.

An appropriate fraction of produced samples was selected by sieving, according to the particle size determined by the experimental design. Moisture content was adjusted after moisture analysis based on weight loss on drying (Mettler LP16 moisture analyser, Mettler-Toledo, Zaventem, Belgium). The proper amount of purified water was added to a fluid bed dryer Mini-Glatt 4 (Glatt GmbH, Binzen, Germany), applying the top spray method. Then, moisture and particle size analysis was performed again.

The glidant/lubricant mixture was finally added, and 5 min of blending was applied. 

The glidant used in our experiments was talc, a commonly used glidant, and the lubricant was magnesium stearate, which is also described as having glidant properties. The amount of both glidant and lubricant was constant at 5% of the total mass, and was used in different ratios, according to Table 2. Each examined sample had a total mass of 100 g.

### 2.2. Experimental Setup for Powder Flow Measurements

Merging the theory of the static and dynamic angle of repose measurement, new equipment was created. The main body of the equipment was a regular acid-resistant stainless steel funnel described in Ph.Eur. 6/2.9.16., measuring 100 ± 0.01 mm top diameter, 100 ± 0.01 mm length, and with an outflow opening of 10 ± 0.01 mm cut perpendicular to the axis of symmetry. The funnel was mounted above a plexiglass plane. The distance between the plexiglass and the nozzle was 60 mm.

A regular CMOS camera (Logitech QuickCam V-UAP9, 25 frames per second, 128 × 160 pixel resolution) was applied as the detector compartment was placed 200 mm from the base point of the nozzle, so its optical axis was on the plane of the plexiglass towards the nozzle. The whole device was placed in front of a black pane using forward illumination to enhance the image analysis.

### 2.3. Image Analysis and Evaluation

The measurement was done by loading the funnel with 100 g of the sample and recording the contour of particle flow using the digital CMOS camera (Appendix A). The forming of a particle pile in 3D was thus converted to 2D. In order to retrieve suitable data for quantitative analysis, three preliminary steps were taken.

The first step was extracting all captured frames from the recorded material. The second step was the slicing of the frames vertically into 32 equal pieces.

Choosing the right slice for the analysis was a crucial point. Middle slices of the captured frames were technically unfeasible since the bulk flow occurred there. Terminal slices were unsuitable as well because there was no event in the first seconds there. Thus, only one slice was chosen, as close to the middle of the frame as possible, so that the bulk flow did not disturb the analysis (Figure 1).

The third step was the digital image enhancement and measurement using Carl Zeiss™ AxioVision Rel. 4.8.2 software. The image analysis of the slices consisted of the allocation of bright space (representing the particles) and dark space in the image (representing the black background). Measured areas of bright spaces of single slices were used for the statistical evaluation to characterize the particle flow.

Software TableCurve^®^ 2D v5.01 (Systat Software Inc., London, UK) was used to evaluate the data set. A graph was created plotting the measured areas in the function of time, which resulted in a particle flow curve. Avalanche behavior could be observed, forming sine wave-like patterns. Several curve characteristics were measured, such as wavelength, amplitude, and count of avalanches, which were both averaged and observed in function of time. Non-linear fitting was performed using TableCurve^®^’s power law function (2).

Flow time was measured from the recorded material, and static angle-of-repose was directly calculated from the last frame, averaging the angles on both sides of the pile.

### 2.4. Sphericity Determination

Sphericity (Ψ) of granules and pellets was determined via microscopic examination at 160× magnification (Carl Zeiss Axio Imager A1 Microscope, Carl Zeiss MicroImaging GmbH, Jena, Germany) using a 5 megapixel camera (Carl Zeiss AxioCam MRc 5, (Carl Zeiss MicroImaging GmbH, Jena, Germany), examining n = 50 particles from both granules and pellets. Digital image analysis was carried out using Carl Zeiss Axio Vision Rel. 4.7 software and Formula (2):(2)$$\psi =4\pi \frac{A_{f}}{P_{Cr}^{2}}$$
where *A_f_* is the 2D projected area of the randomly selected particles, and *P_Cr_* is their Crofton perimeter.

According to the sphericity calculation, a perfectly round-shaped particle’s sphericity index is 1.

## 3. Results and Discussion

The flow curve is based on capturing real-time image data of a projected conic section (described in Section 2.3), which changes rapidly due to particle movement. The principle of this method contains the simplicity of a basic cone-growth model and the complexity of self-organized criticality. Cone growth could be characterized by the growth of the volume of the cone filled with particles. In the case of free or easy-flowing powders, the measured flow curve is proportional to the changing height of the cone, so finding the relationship between the growing height and the volume of the cone-shaped particle pile helps to understand the theoretical background of the non-linear fitting. Figure 2 represents a flow curve measured by our method and the non-linear fitting with the power law function, Equation (3):(3)$${y=a+bt}^{c}$$
where ‘*y*’ is the measured area (Section 2.3), ‘*t*’ is the time, ‘*a*’ and ‘*b*’ are constants, and ‘*c*’ is the scaling exponent. 

Assuming the static angle of repose to be nearly constant, radius ‘*r*’ can be expressed as (4):(4)$$r=\frac{h}{tg(\alpha )}$$
where ‘*h*’ is the height of the cone, *α* is the static angle of repose.

Substituting ’*r*’ in the formula for the volume of the circular cone, it becomes (5):(5)$$V=\frac{1}{3}\pi \frac{h^{3}}{{tg}^{2}(\alpha )}$$

The measured curve proportional to the height of the cone can be expressed as (6):(6)$$h=\sqrt[{3}]{\frac{3V{tg}^{2}(\alpha )}{\pi }}$$

Rearranging the formula, the outline of a power law function can be discovered in expression (7):(7)$$h={\left(\frac{3}{\pi }\right)}^{\frac{1}{3}}\cdot {tg}^{\frac{2}{3}}(\alpha )\cdot V^{\frac{1}{3}}$$

Comparing Formulas (3) and (7) and assuming that the calculated *‘y*’ value and the measured ‘*h*’ height, as well as the growing ‘*V*’ volume of the cone and ‘*t*’ time, are in linear relationship respectively (y∝h, t∝V), we can hypothesize (8):(8)$$b\propto {\left(\frac{3}{\pi }\right)}^{\frac{1}{3}}\cdot tg^{\frac{2}{3}}(\alpha );\;c=\frac{1}{3}$$

Value ‘*a*’ refers to the intercept, which is constant according to Equation (3), but we could hypothesize that it substitutes an asymptotic notation ‘*o(x)k*’ or deviation term ‘*ε*’. It can also represent the avalanche behavior of the pile.

Value ‘*b*’ is proportional to the static angle of repose and the height of the pile; however, if the observation is limited to a definite interval of time, the relationship for value ‘*b*’ described above (8) is not necessarily valid. Moreover, it is often in a negative relationship with the static angle of repose. This phenomenon could be explained with the “flow through an orifice” method used in the experiments, since the flowability of particles determines the flow rate through the orifice. The same weight of examined particles with good flowability produces lower piles than particles with poor flowability, while within a definite interval of time, particles with good flowability produce higher piles than particles with poor flowability. 

Value ‘*c*’ originates from the formula for the volume of the cone. The higher the value ‘*c*’ is, the more cylindrical the pile becomes. In practice, value ‘c’ refers to the growth rate of the pile.

Evaluation of the flow curve (Figure 2) was very complex, since with the number of parameters able to be obtained, such as pile growth rate, avalanche behavior (count, amplitude, frequency), and using different functions fitting the curve, several other parameters were able to characterize the flow property.

### 3.1. Static Angle of Repose (SAOR) and Flow Time (FT)

Control measurements were carried out using the classic static angle of repose measurement method and observing flow time. Table 3 represents the results.

According to the evaluation of the central composite design, the linear 2 factor interaction model is significant (*p* < 0.0001) for the data of static angle of repose. In this measurement, factor x_4_ (sphericity, *p* < 0.0001); factor x_1_ (particle size, *p* = 0.0082); factor x_2_ (moisture content, *p* = 0.0485); and interactions ‘x_1_x_3_’ (*p* = 0.0151) and ‘x_3_x_4_’ (*p* = 0.0161) were significant, where ‘x_3_’ is the glidant/lubricant ratio. Figure 3, Figure 4, Figure 5 and Figure 6 represent response surfaces of the static angle of repose measurement. Equation (9) is the mathematical formula of the response surface.
SAOR = 34.41 + 1.45x_1_ − 1.01 x_2_ − 0.92 x_3_ − 2.46 x_4_ + 0.19x_1_x_2_ − 1.84x_1_x_3_ − 0.55x_1_x_4_ + 0.19x_2_x_3_ + 0.081x_2_x_4_ − 0.91x_3_x_4_(9)

The quadratic model (Equation (10)) was significant (*p* < 0.0001) for the flow time of the samples. Factors x_1_ (particle size, *p* = 0.0001); x_3_ (glidant/lubricant ratio, *p* = 0.0445); x_4_ (sphericity, *p* < 0.0001), x_3_x_4_ (*p* = 0.0016); and x_12_ (*p* < 0.0001) were significant. Figure 7, Figure 8, Figure 9 and Figure 10 represent the response surfaces of the model.
FT = 14.78 − 0.98x_1_ − 0.052x_2_ − 0.36x_3_ − 4.01D x_4_ + 0.31x_1_x_2_ + 0.44x_1_x_3_ + 0.18x_1_x_4_ − 0.44x_2_x_3_ − 0.13x_2_x_4_ − 0.49x_3_x_4_ + 1.17x_1_^2^ − 0.13x_2_^2^ − 0.15x_3_^2^(10)

Sphericity and particle size were the most significant factors affecting both measured parameters; moisture content affected only the static angle of repose, and the glidant/lubricant ratio affected only the flow time. Figure 11 demonstrates the difference between granules and pellets on sample flow time.

### 3.2. Avalanche Behavior

Avalanche behavior was examined by evaluating the flow curve, counting the number of avalanches, and determining their amplitude. Results are summarized in Table 4, and the heat map of the model is presented in Figure 12 and Figure 13.

According to the response surfaces, none of the factors significantly affect the avalanches’ amplitude; on the other hand, wavelength mean and avalanche count could be used to characterize flow patterns. The reduced response surface linear model was adequate for the wavelength (Equation (11), *p* = 0.0002) and count of avalanches (Equation (12), *p* = 0.0066). In both cases, only factor x_4_ (sphericity) was significant (*p* < 0.0001).
Avalanche wavelength = 21.15 − 1.82x_4_(11)
Avalanche count = 25.16 + 2.71x_4_(12)

According to the equations above, avalanche wavelength was longer at granules and shorter at pellets, and avalanche count was rare at granules and frequent at pellets.

### 3.3. Non-Linear Fitting of Flow Curve

Function (3) was used for the curve fitting of the flow graph. Parameters ‘a’, ‘b’, and ‘c’, ‘a + b’, and the area under the flow curve (AUFC) were determined and summarized in Table 5; response surfaces of adequate models are represented in Figure 14, Figure 15, Figure 16, Figure 17, Figure 18 and Figure 19.

Table 6 contains the correlation of all examined parameters, including parameters of empirical methods. ANOVA analysis of Predicted R-squared was negative for parameter ‘a’, which means the overall mean of ‘a’ is a better predictor for the model. The reduced quadratic response surface model was adequate for parameter ‘b’ (Equation (13), *p* < 0.0001), in which factor x_1_ (particle size, *p* = 0.0002) and factor x_4_ (sphericity, *p* < 0.0001); interaction x_3_x_4_ (*p* = 0.0025); and squared factors x_1_^2^ (0.0001) and x_2_^2^ (*p* = 0.0394) were significant.
Parameter ‘b’ = 28.81 + 2.58x_1_ − 0.46x_2_ + 0.50x_3_ + 2.50x_4_ − 2.00x1x_2_ + 2.07x_1_x_3_ + 1.90x_3_x_4_ − 3.12x_1_^2^ + 1.33x_2_^2^(13)

The reduced quadratic model was adequate for parameter ’c’ (Equation (14), *p* = 0.0001), where factor x_1_ (particle size, *p* = 0.0092); factor x_4_ (sphericity, *p* < 0.0001); interaction x_3_x_4_ (*p* = 0.0108); and x_1_^2^ (*p* = 0.0007) were significant.
Parameter ‘c’ = 0.41 − 0.010x_1_ + 7.372 × 10^−4^x_2_ − 2.088 × 10^−3^x_3_ − 0.015x_4_ + 6.848 × 10^−3^x_1_x_2_ − 0.013x_1_x_3_ − 9.972 × 10^−3^x_3_x_4_ + 0.017x_1_^2^ − 7.294 × 10^−3^x_2_^2^(14)

An adequate model (Equation (15)) for parameter ’a + b’ was linear with *p* = 0.0001. In this case, factor x_2_ (moisture content, *p* = 0.03) and factor x_4_ (sphericity, *p* < 0.0001) were significant.
Parameter ‘a + b’ = −11.66 − 0.79x_1_ + 1.32x_2_ − 0.33x_3_ + 2.70x_4_(15)

The reduced quadratic model (Equation (16)) seemed to be adequate for the area under the flow curve (*p* < 0.0001). Factor x_1_ (particle size, *p* = 0.0003); factor x_4_ (sphericity, *p* < 0.0001); and squared factor x_1_^2^ were significant in this case.
Parameter ‘AUFC’ = 26792.27 + 988.31x_1_ + 805.76x_4_ − 1282.55x_1_^2^(16)

According to the adequate models at time-limited measurements, parameter ‘b’ reaches a maximum at around 900 μm particle size, then slightly decreases. At granules, parameter ‘b’ decreases, and at pellets, it increases.

Parameter ‘c’ originated from the volume of the theoretical cone of the pile and is in close negative correlation with parameter ‘b’. 

Parameter ‘a + b’, which is probably a combination of the avalanche behavior and the angle of repose/avalanche, is the only parameter of the non-linear fitting affected significantly by the moisture content. As the moisture increases, ‘a + b’ increases. The increasing sphericity increases its value as well.

The parameter ‘AUFC’ is negatively correlated with the flow time. After AUFC increases, it reaches a maximum value of 900 μm and decreases as the particle size increases. Granules decrease, and pellets increase in value. The results of our examinations can provide a good characterization and a suitable model of how solid particles will flow during manufacturing processes, and help to determine the appropriate operating parameters for production.

## 4. Conclusions

The static angle of repose measurement combined with real-time image analysis offers new possibilities in powder flowability determination techniques. This method can detect avalanche behavior, static angle of repose, and flow time simultaneously. Creating a geometrical background of the cone-growth model for the measurements, the coefficients of non-linear fitting could be associated with definite geometrical parameters. By examining the effect of particle size; moisture content; glidant/lubricant ratio; and the role of pellets or granules (sphericity), it could be established that sphericity and particle size are the most significant factors affecting flowability measurements. However, the cone-growth model could also determine the effect of other less important factors like electrostatic charge, the chemical nature of material, temperature, and powder density. Parameters of avalanche behavior were significantly affected only by the sphericity; thus, in our experiments, self-organized criticality during the toppling of particles revealed only a small part of the hidden relations between examined factors and the flowability.

## Figures and Tables

**Figure 1 pharmaceutics-16-00405-f001:**
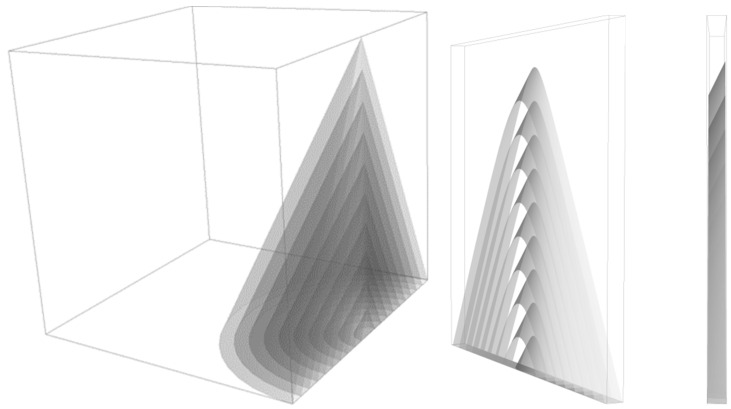
Image analysis of particle flow.

**Figure 2 pharmaceutics-16-00405-f002:**
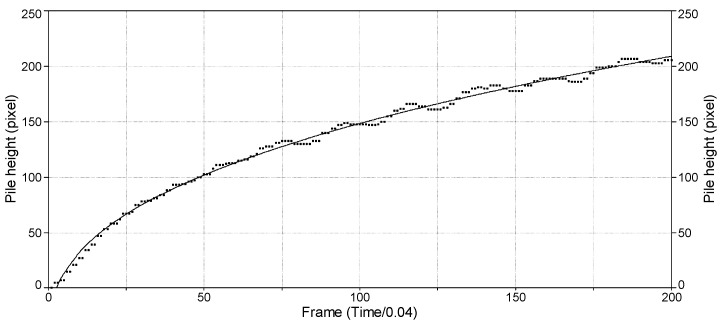
Flow curve of a sample and its non-linear fitting.

**Figure 3 pharmaceutics-16-00405-f003:**
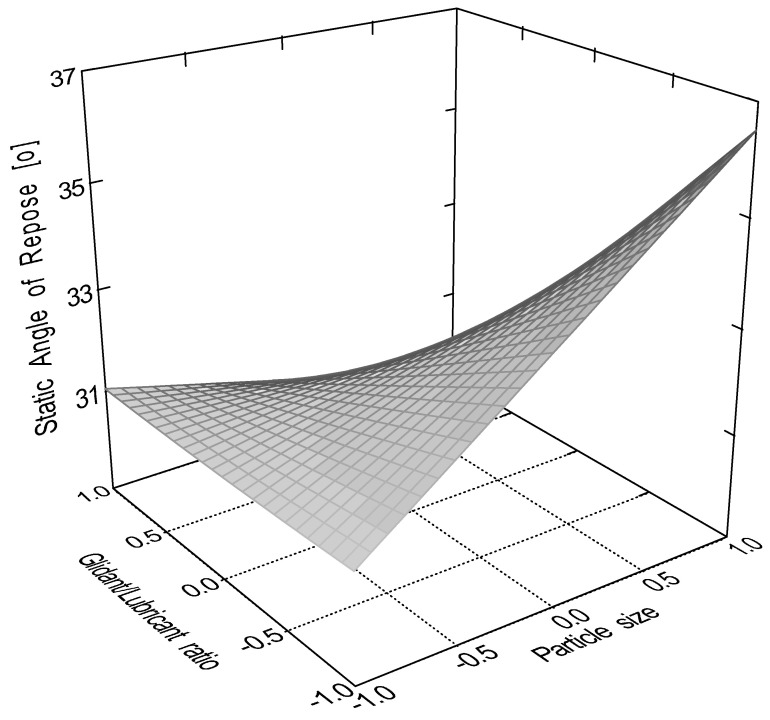
Effect of the glidant/lubricant ratio and particle size of pellets on the static angle of repose measurement.

**Figure 4 pharmaceutics-16-00405-f004:**
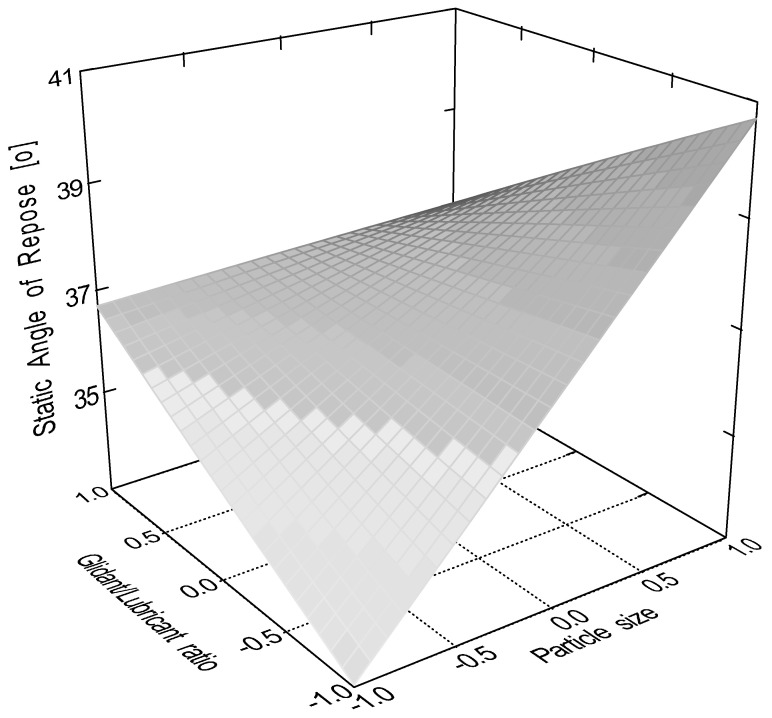
Effect of the glidant/lubricant ratio and particle size of granules on the static angle of repose measurement.

**Figure 5 pharmaceutics-16-00405-f005:**
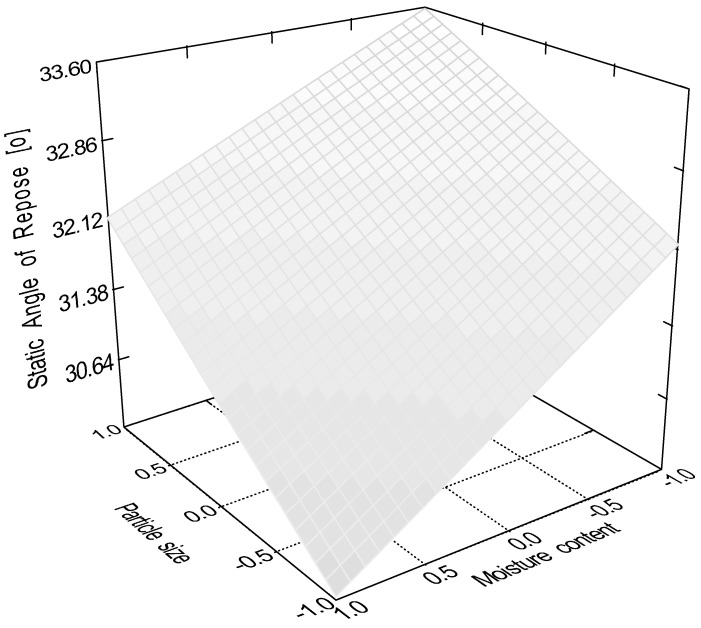
Effect of the moisture content and the particle size of pellets on the static angle of repose measurement.

**Figure 6 pharmaceutics-16-00405-f006:**
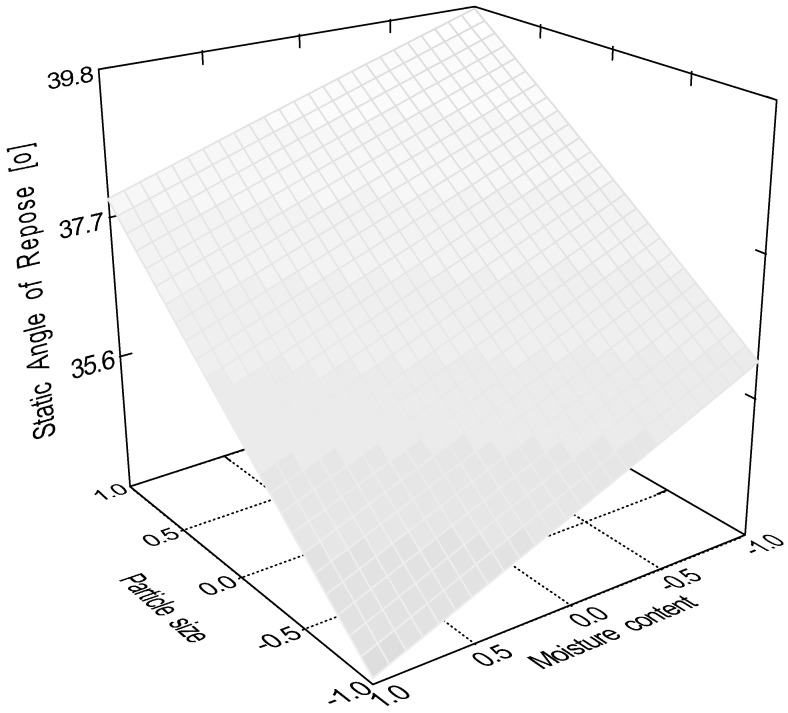
Effect of the moisture content and the particle size of granules on the static angle of repose measurement.

**Figure 7 pharmaceutics-16-00405-f007:**
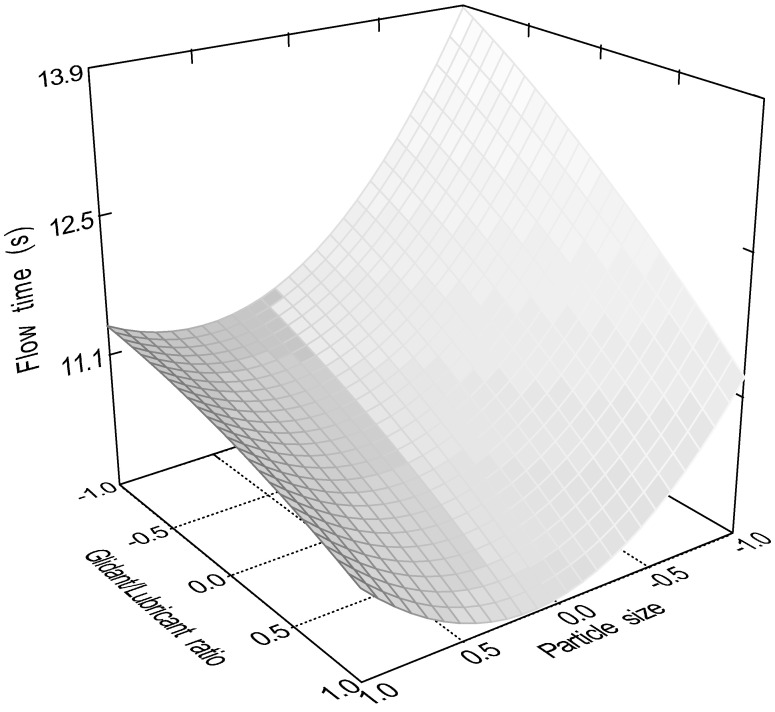
Effect of the glidant/lubricant ratio and the particle size of pellets on the flow time measurement.

**Figure 8 pharmaceutics-16-00405-f008:**
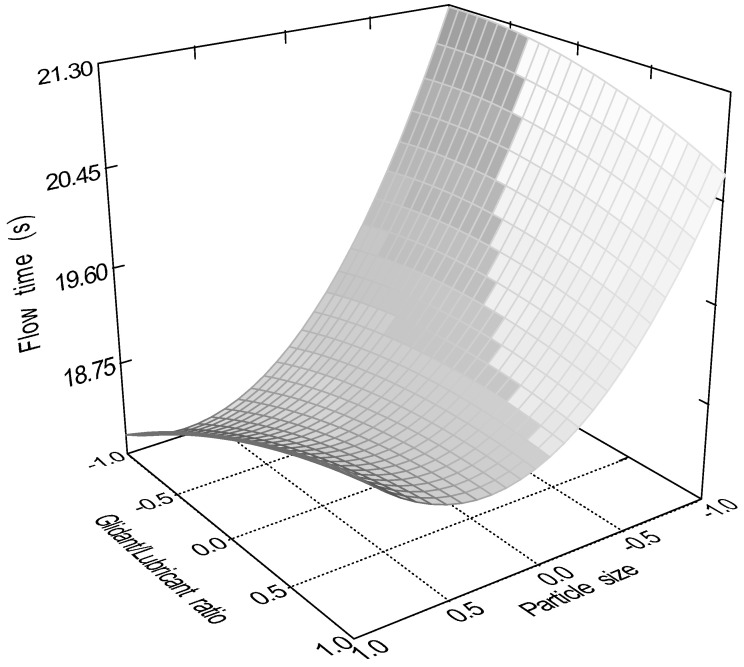
Effect of the glidant/lubricant ratio and the particle size of granules on the flow time measurement.

**Figure 9 pharmaceutics-16-00405-f009:**
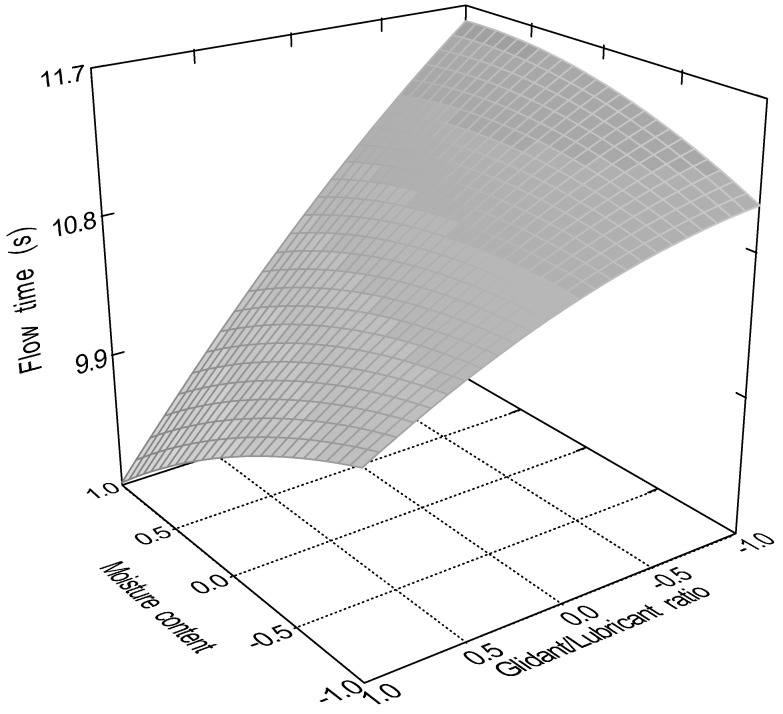
Effect of the moisture content and the particle size of pellets on the flow time measurement.

**Figure 10 pharmaceutics-16-00405-f010:**
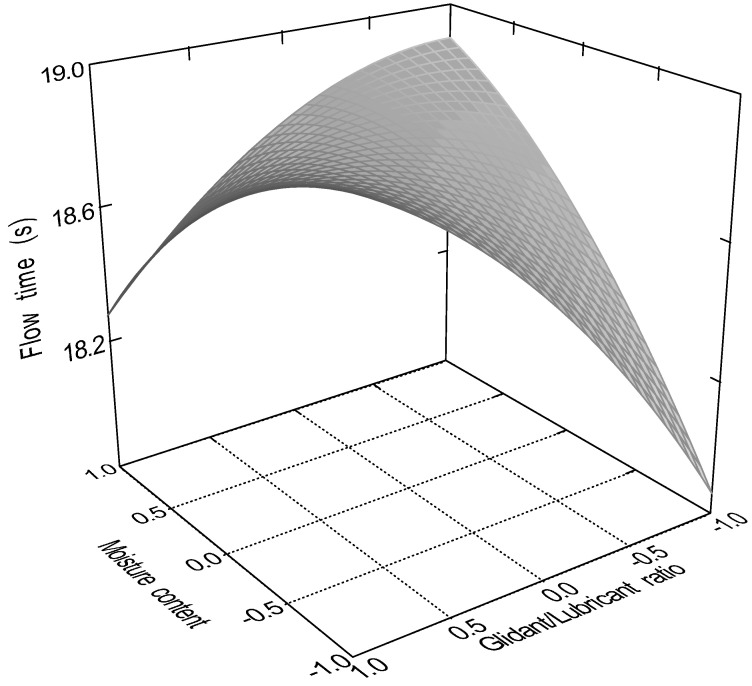
Effect of the moisture content and the particle size of granules on the flow time measurement.

**Figure 11 pharmaceutics-16-00405-f011:**
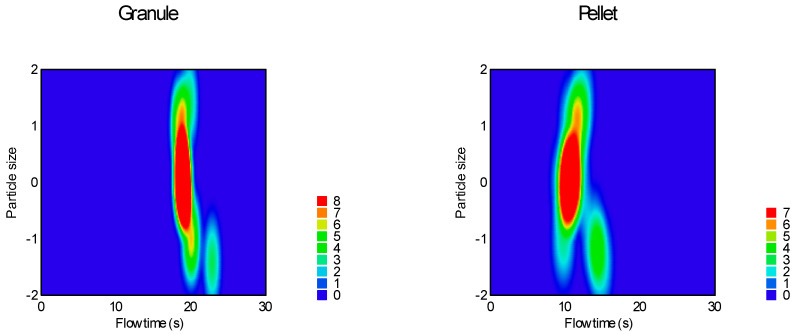
Heat map of effect of particle size and sphericity on flow time.

**Figure 12 pharmaceutics-16-00405-f012:**
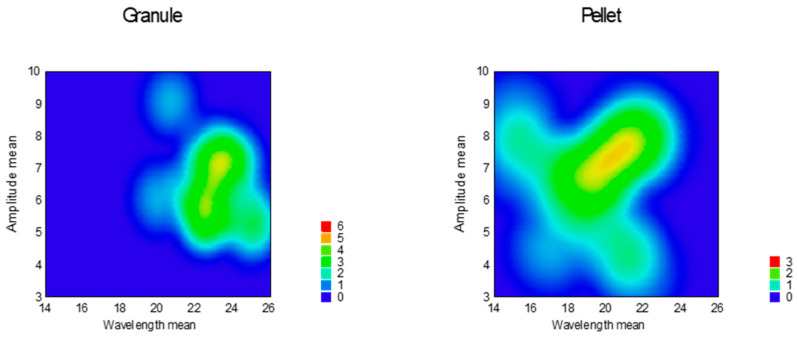
Heat map of effect of sphericity on avalanche amplitude and wavelength.

**Figure 13 pharmaceutics-16-00405-f013:**
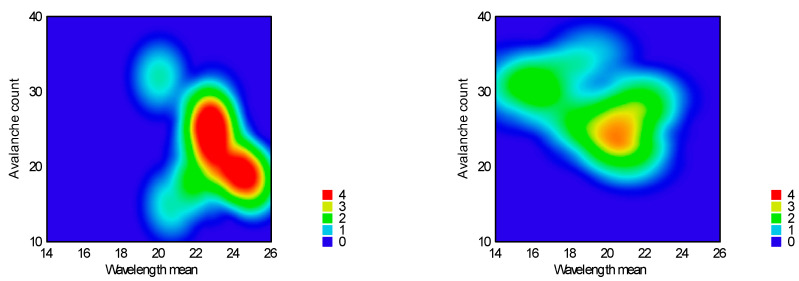
Heat map of effect of sphericity on avalanche count and wavelength.

**Figure 14 pharmaceutics-16-00405-f014:**
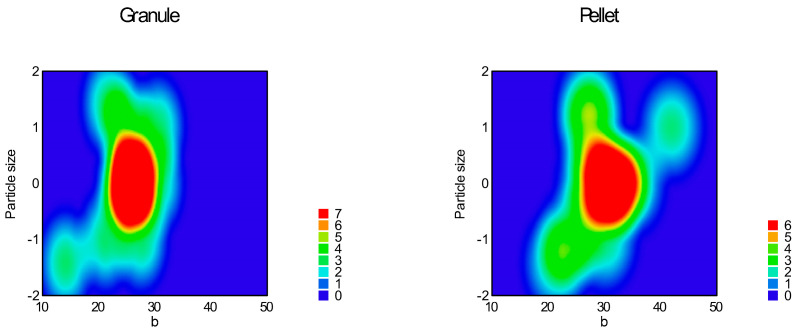
Heat map of effect of particle size and sphericity on parameter ‘b’.

**Figure 15 pharmaceutics-16-00405-f015:**
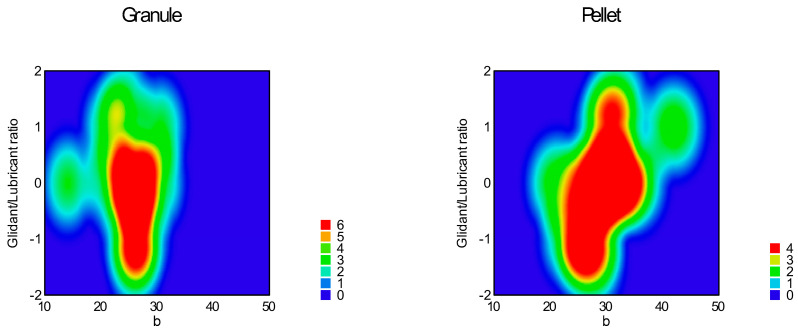
Heat map of effect of glidant/lubricant ratio and sphericity on parameter ‘b’.

**Figure 16 pharmaceutics-16-00405-f016:**
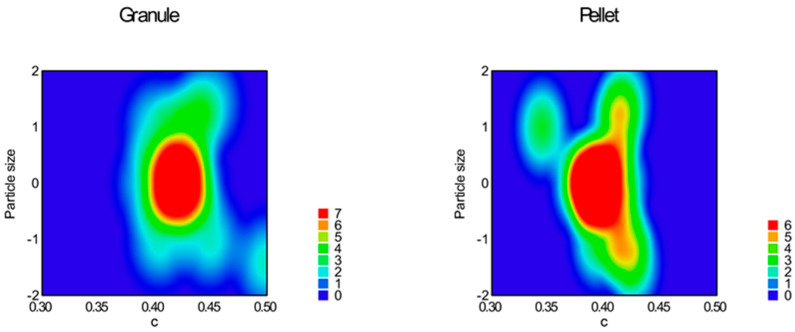
Heat map of effect of particle size and sphericity on parameter ‘c’.

**Figure 17 pharmaceutics-16-00405-f017:**
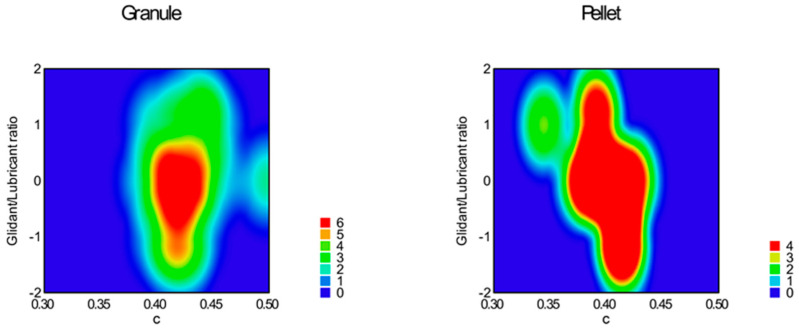
Heat map of effect of glidant/lubricant ratio and sphericity on parameter ‘c’.

**Figure 18 pharmaceutics-16-00405-f018:**
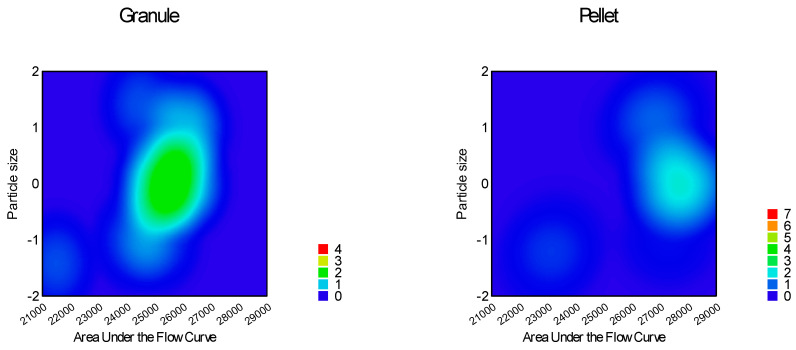
Heat map of effect of particle size and sphericity on parameter ‘AUFC’.

**Figure 19 pharmaceutics-16-00405-f019:**
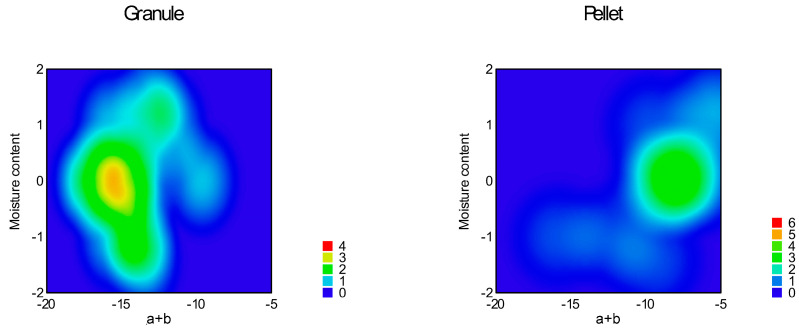
Heat map of effect of moisture content and sphericity on parameter ‘a + b’.

**Table 1 pharmaceutics-16-00405-t001:** Experimental design.

	Factor x_1_	Factor x_2_	Factor x_3_	Factor x_4_
Sample	Particle Size	Moisture Content	Glidant/Lubricant Content	Granule (−1) Pellet (1)
1	1100.0	3.0	−1.0	−1
2	1100.0	1.0	1.0	−1
3	350.0	3.0	1.0	−1
4	350.0	1.0	−1.0	−1
5	725.0	2.0	0.0	−1
6	194.7	2.0	0.0	−1
7	1255.3	2.0	0.0	−1
8	725.0	0.6	0.0	−1
9	725.0	3.4	0.0	−1
10	725.0	2.0	−1.4	−1
11	725.0	2.0	1.4	−1
12	725.0	2.0	0.0	−1
13	1100.0	3.0	−1.0	1
14	1100.0	1.0	1.0	1
15	350.0	3.0	1.0	1
16	350.0	1.0	−1.0	1
17	725.0	2.0	0.0	1
18	194.7	2.0	0.0	1
19	1255.3	2.0	0.0	1
20	725.0	0.6	0.0	1
21	725.0	3.4	0.0	1
22	725.0	2.0	−1.4	1
23	725.0	2.0	1.4	1
24	725.0	2.0	0.0	1

**Table 2 pharmaceutics-16-00405-t002:** Glidant/lubricant content of granules.

Factors Coded Value	−1.4	−1.0	0.0	1.0	1.4
Talc content %	0	1.25	2.5	3.75	5
Mg-stearate content %	5	3.75	2.5	1.25	0

**Table 3 pharmaceutics-16-00405-t003:** Results of static angle of repose and flow time measurement.

Sample	SAOR (°)	FT (s)
1	38.77	18.31
2	37.14	18.62
3	34.78	19.75
4	34.02	20.33
5	39.77	19.15
6	34.28	22.83
7	40.02	19.78
8	39.10	19.10
9	36.45	18.95
10	36.10	18.84
11	36.08	18.88
12	35.84	18.62
13	35.54	11.72
14	29.38	10.70
15	30.02	9.76
16	31.93	13.80
17	32.75	10.49
18	31.38	14.79
19	33.85	12.32
20	32.29	10.70
21	29.25	10.55
22	34.65	11.73
23	29.44	9.64
24	32.90	10.60

**Table 4 pharmaceutics-16-00405-t004:** Summarized results of avalanche behavior.

Sample	Avalanche Amplitude (Pixel)	Avalanche Wavelength (Time/0.04)	Avalanche Count
1	5.47	25.17	18.00
2	7.00	23.29	24.00
3	7.54	22.73	26.00
4	5.39	22.86	28.00
5	5.38	22.27	26.00
6	6.07	20.03	32.00
7	5.09	24.94	18.00
8	7.43	23.86	21.00
9	6.98	24.37	19.00
10	5.71	22.61	18.00
11	6.65	22.05	21.00
12	9.07	20.67	15.00
13	6.08	19.11	35.00
14	4.54	17.00	32.00
15	3.80	21.56	25.00
16	7.51	21.45	29.00
17	7.37	20.00	26.00
18	7.96	22.24	29.00
19	6.94	19.08	24.00
20	6.62	17.96	27.00
21	8.52	15.19	32.00
22	7.19	15.55	29.00
23	4.79	20.64	22.00
24	8.29	21.33	21.00

**Table 5 pharmaceutics-16-00405-t005:** Summarized parameters of non-linear fitting of flow curves.

Sample	a	b	c	a + b	AUFC
1	−36.85	25.05	0.43	−11.79	26,031.05
2	−46.14	30.57	0.40	−15.57	26,326.58
3	−36.95	21.26	0.45	−15.70	24,242.23
4	−41.38	27.56	0.40	−13.81	25,211.94
5	−45.18	30.37	0.39	−14.81	26,181.58
6	−23.75	14.11	0.50	−9.64	21,505.26
7	−35.44	21.73	0.45	−13.71	24,494.15
8	−39.72	26.23	0.42	−13.50	26,183.06
9	−39.30	26.23	0.42	−13.07	25,623.30
10	−41.21	26.06	0.42	−15.16	25,164.99
11	−41.11	24.32	0.43	−16.79	25,313.01
12	−42.45	25.18	0.43	−17.27	24,918.38
13	−36.97	27.37	0.41	−9.61	26,429.78
14	−57.80	41.95	0.34	−15.85	26,903.19
15	−35.86	30.09	0.39	−5.77	27,293.46
16	−36.16	23.92	0.42	−12.23	23,353.74
17	−38.38	29.08	0.40	−9.29	27,599.63
18	−28.46	21.37	0.43	−7.09	22,905.91
19	−35.44	27.27	0.42	−8.17	27,146.24
20	−44.48	35.10	0.38	−9.39	28,408.06
21	−37.71	32.49	0.39	−5.22	27,732.51
22	−35.58	27.30	0.42	−8.28	27,499.81
23	−38.33	31.85	0.39	−6.48	27,690.18
24	−39.49	30.95	0.39	−8.54	27,571.28

**Table 6 pharmaceutics-16-00405-t006:** Correlation of parameters of non-linear fitting and empirical measurements.

		a	b	c	AUFC	a + b	av. wav.	av. amp.	av. count	SAOR	FT
a	Pearson corr.	1	−0.808 **	0.749 **	−0.499 *	0.515 *	0.12	0.33	0.096	0.01	0.16
	Sig.		0	0	0.013	0.01	0.576	0.115	0.654	0.964	0.456
b	Pearson corr.	−0.808 **	1	−0.970 **	0.808 **	0.089	−0.440 *	−0.495 *	0.151	−0.413 *	−0.651 **
	Sig.	0		0	0	0.679	0.032	0.014	0.481	0.045	0.001
c	Pearson corr.	0.749 **	−0.970 **	1	−0.764 **	−0.145	0.39	0.513 *	−0.171	0.456 *	0.690 **
	Sig.	0	0		0	0.499	0.06	0.01	0.426	0.025	0
AUFC	Pearson corr.	−0.499 *	0.808 **	−0.764 **	1	0.333	−0.417 *	−0.271	−0.029	−0.247	−0.701 **
	Sig.	0.013	0	0		0.111	0.043	0.2	0.893	0.245	0
a + b	Pearson corr.	0.515 *	0.089	−0.145	0.333	1	−0.437 *	−0.162	0.383	−0.585 **	−0.677 **
	Sig.	0.01	0.679	0.499	0.111		0.033	0.449	0.065	0.003	0
av. wav.	Pearson corr.	0.12	−0.440 *	0.39	−0.417 *	−0.437 *	1	0.048	−0.622 **	0.623 **	0.646 **
	Sig.	0.576	0.032	0.06	0.043	0.033		0.824	0.001	0.001	0.001
av. amp.	Pearson corr.	0.33	−0.495 *	0.513 *	−0.271	−0.162	0.048	1	−0.203	0.408 *	0.36
	Sig.	0.115	0.014	0.01	0.2	0.449	0.824		0.341	0.048	0.084
av. count	Pearson corr.	0.096	0.151	−0.171	−0.029	0.383	−0.622 **	−0.203	1	−0.501 *	−0.353
	Sig.	0.654	0.481	0.426	0.893	0.065	0.001	0.341		0.013	0.091
SAOR	Pearson corr.	0.01	−0.413 *	0.456 *	−0.247	−0.585 **	0.623 **	0.408 *	−0.501 *	1	0.728 **
	Sig.	0.964	0.045	0.025	0.245	0.003	0.001	0.048	0.013		0
FT	Pearson corr.	0.16	−0.651 **	0.690 **	−0.701 **	−0.677 **	0.646 **	0.36	−0.353	0.728 **	1
	Sig.	0.456	0.001	0	0	0	0.001	0.084	0.091	0	

** Correlation is significant at the 0.01 level (2-tailed); * Correlation is significant at the 0.05 level (2-tailed).

## Data Availability

The data that support the findings of this study are available from the corresponding author S.P.

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
