# Peer review of "Real-Time Cone-Growth Model for Determination of Pharmaceutical Powder Flow Properties"

_pharmaceutics, 2024, doi:10.3390/pharmaceutics16030405_

Round 1

Reviewer 1 Report

Comments and Suggestions for Authors

The manuscript titled “Real-time cone-growth model for determination of pharmaceutical powder flow properties” is interesting and well discussed. The authors have discussed the new techniques as well as the empirical studies applicability. This study could be valuable to the researchers in industry and academia focused on processing granules and pellets dosage forms which are intrinsically complex in terms of manufacturing.

Author Response

Thank you for your time and positive opinion about our research.

Reviewer 2 Report

Comments and Suggestions for Authors

Overall, the work presented by authors is interested adn has merit to be published in the journal. However, there are few points that need revision before publication. 

1. The introduction can be extended to understnad the rational behind the work carried out .

2. The methodology requires to be expanded in more detail, especially the design of experiments adn how these parameters have been selected. 

3. The reuslts should be explained more in depth to actually understand the impact on the final dosage form. 

4. The discussion is poor, authors should compare their results with those obtained by other researchers. For example, Can you extrapolate these values to industrial manufacturing? Wich characteristi for flow are better to ensure a suitable flow?Regarding granules, which one will flow better to fill capsules? See paper: Optimising the in vitro and in vivo performance of oral cocrystal formulations via spray coating

Comments on the Quality of English Language

English spelling has to be checked. 

Author Response

Thank you for your time and positive evaluation of our work. We are mostly grateful for constructive highlight of week points in our manuscript. We have addressed all the points you have highlighted. Hope our corrections and responses will meet the requirement for publication in Pharmaceutics.

Please find our responses blow:

  1. The introduction can be extended to understand the rationale behind the work carried out.

Response:

The introduction has been extended to clarify the rational of this research:

 (lines 31-35 in the revised manuscript)

“Developments in the measurement of powder flow characteristics need to take into account the way in which powder filling, tableting or capsule filling machines operate during the production of pharmaceuticals. In this way, it can be well modelled whether the flowability of powders will be adequate during drug production, which is also in line with the principles of QbD.”

  1. The methodology requires to be expanded in more detail, especially the design of experiments and how these parameters have been selected.

Response:

The methodology has been expanded in more details: (Lines 342-348 in the revised manuscript)

“We have set up our experimental design by grounding our parameter selection based on robust scientific literature that identifies particle size, shape, moisture, inter-particle forces and added glidant excipients as critical determinants of powder flowability. Our methodology integrates a central composite experimental design to systematically explore the impact of these parameters on flow properties. This approach allows us to delineate the influence of each variable and their interactions, as supported by significant empirical evidence.”

  1. The reuslts should be explained more in depth to actually understand the impact on the final dosage form.

Response:

The results are good for the characterization of the flowability, they do not influence the examined formlations in any way. Tthe usage of the parameters detrmined with our methode are suitable for definition of appropriate manufactoring parameters. It is critical to highlight the significance of material flow properties in pharmaceutical manufacturing. Our findings, which focus on characterizing the flowability of the material, have direct implications for the production process, particularly concerning the uniformity of dosage forms. Poor flow properties can significantly compromise the weight uniformity of the dosage forms, leading to variations that may affect both the product's quality and its therapeutic efficacy.

  1. The discussion is poor, authors should compare their results with those obtained by other researchers. For example:

4.1. Can you extrapolate these values to industrial manufacturing?

Response:

We have added a new part to the results and discussion:

(lines 319-322 in the revised manuscript)

The results of our examinations can provide a good characterization and a suitable model of how solid particles will flow during manufacturing processes and help to determine the appropriate operating parameters for production.

4.2. Wich characteristi for flow are better to ensure a suitable flow? 1. mondat (depends on formulation but the parameters obtained by our methode can provide a sutible model to select optimal parameters for the desired flow)

Response:

It depends on formulation, but the flow describing parameters obtained by our methode can provide a sutible model to select optimal characterisctics for the desired flow.

            4.3. Regarding granules, which one will flow better to fill capsules?  See paper: Optimising the in vitro and in vivo performance of oral cocrystal formulations via spray coating

Response:

We examined well-known powder formulations in order to establish a new method for characterization o powder flowability. It was not in the scope of this work to evaluate the quality of flow for specific application like capsule filing. The parameters obtained by our method can be used to evaluate the quality of the flow for specific application.

Comments on the Quality of English Language

English spelling has to be checked.

Response:

English Language has been checked and edited by Grammarly.

Reviewer 3 Report

Comments and Suggestions for Authors

In this paper, the authors tended to establish a cone-growth model for determination of flow properties of pharmaceutical powders. From the practical point view, the study has limited novelty. However, it is good to see an exploration of measuring static angles of repose and dynamic responses in creating a powder cone. There are a few major concerns and detailed suggestions shown as follows.

1, The abstract needs to be improved by any achievements or conclusions drawn by the study.

2, Introduction: it must be improved and more references should be included. The introduction does not cover the research gaps in the literatures, especially the proposed study and the novelties of the current study.

3, Line 33: formatting, remove the space in between the paragraphs.   

4, Line 34: ‘applying basic methods’, what are the basic methods?

5, Line 42: ‘Particles below 50μm practically have no flow due to van der Waals forces.’ The statement is not true. There are many powders which are smaller than 50mm and can flow. However, the powders (<50mm) can be cohesive or very cohesive. For the powders, flow will not be an issue, but it hard to handle.

6, Line 45: reference and citation.

7, Line 48-49: Low moisture content could contribute the generation of electrostatic charging but it cannot be an essential reason or the only reason. Charging of materials is mainly a characteristic of the materials. 

8, Line 51: references.  

9, Line 54: is the equation correct?

10, Line 63: ‘Spherical and smooth particles have sufficient flow.’, it is not accurate. If particle size is very small, flow can be an issue.   

11, Line 68: reference.

12, Line 79: how is the ‘projected conic section’ created? If the materials and methods are presented in the later section, here it should be mentioned.

13, Line 83: ‘Measured flow curve is proportional to the changing height of the cone.’ This statement is correct only if the powder is free-flowing or easy-flowing.

14, Line 85: It is questionable on the correlation between the powder flow properties and the ‘non-linear fitting’ model proposed here.

15, Line 89: 'y' is the measured area, what is the ‘measured area’?

16, Line 133-136: the statement is not accurate, which depends on many factors.

17, Data in Table 1: In principle, free-flowing materials will have a low angle of repose and a short flow time (discharge time from a funnel). In the table, the data sounds not following the rules especially for the 13-15 samples.

18, Line 203: Sphericity of the particles have been mentioned many times in the manuscript. However, there is not measurement data presented in the manuscript.

19, Table 3: in the data, is the comma used for a dot as decimals?

20, Line 240: it should be ‘the’ area under the flow curve.

21, Line 249: what is the sphericity of the granule and the pellets?

22, Line 314-317: the physical properties of the ingredients should give, also including the granule and the pellets.  

23, Line 402: what are the ‘other less important factors’?

Comments on the Quality of English Language

English needs to improve accordingly. The formatting of the paper is not adequate. 

Author Response

Thank you for your detailed review of our manuscript. We are mostly grateful for constructive highlight of week points in our manuscript. We have addressed all the points you have highlighted. Hope our corrections and responses will meet the requirement for publication in Pharmaceutics.

Please find our responses below:

  1. The abstract needs to be improved by any achievements or conclusions drawn by the study.

Response:

 The abstract has been modified as suggested in the lines 17-22 of the revised manuscript:

“Our findings indicate that sphericity and particle size are the most significant factors influencing the flowability of pharmaceutical multiparticular preparations. Furthermore, the study confirms that the integration of chaos theory and empirical observations in a time-dependent dynamic model provides a comprehensive understanding of particle flow behavior, which is pivotal for optimizing manufacturing processes.”

  1. 2. Introduction: it must be improved and more references should be included. The introduction does not cover the research gaps in the literatures, especially the proposed study and the novelties of the current study.

Response:

We have added new references and sentences to introduction:

(lines 42-44 in the revised manuscript)

(Currently there is no simple method to determine complex properties of powder flow for Process Analytical Technology (PAT), which is why development of methods that can be implemented in PAT system is high importance.”

and (lines 86-89 in the revised manuscript)

„Aim of this work is to integrate the angle of repose measurement method with chaos theory principles, our study explores the flowability of powders through the lens of self-organized criticality, offering new insights into the dynamics that govern particulate systems.”

  1. Line 33: formatting, remove the space in between the paragraphs.

Response:

The space has been removed between paragraphs.

  1. Line 34: ‘applying basic methods’, what are the basic methods?

Response:

The line has been modified , colon : has been replacet with word “ like” to higlight that the these are the basic methodes

“Flow properties of particles could be characterized in a number of ways, applying basic methods: examining static angle of repose, flow through an orifice, compressibility measurements, shearing tests, observing avalanche behavior, and applying powder rheometers.”

Has been replaced by: (Line 47 in the revised manuscript)

“Flow properties of particles could be characterized in a number of ways, applying basic methods like: examining static angle of repose, flow through an orifice, compressibility measurements, shearing tests, observing avalanche behavior, and applying powder rheometers.”

  1. Line 42: ‘Particles below 50μm practically have no flow due to van der Waals forces.’ The statement is not true. There are many powders which are smaller than 50mm and can flow. However, the powders (<50mm) can be cohesive or very cohesive. For the powders, flow will not be an issue, but it hard to handle.

Response:

Statemant:

“Particles below 50μm practically have no flow due to van der Waals forces. An increase in particle size usually increases flowability.”

Has been replaced by: (Lines 52-53 in the revised manuscript)

“Particles below 50μm practically have irregular flow or no flow due to van der Waals forces. An increase in particle size usually increases flowability.”

  1. Line 45: reference and citation.

Response:

The citation has been given as required by the instruction for authors.

  1. Line 48-49: Low moisture content could contribute the generation of electrostatic charging but it cannot be an essential reason or the only reason. Charging of materials is mainly a characteristic of the materials.

Response:

“Moisture content can affect flow properties too due to forces usually induced by surface tension. Low moisture content can lead to the development of electrostatic charges and inhibits flow [9].”

Has been replaced by: (Lines 56-58 in the revised manuscript)

“Moisture content can affect flow properties too due to forces usually induced by surface tension. Depending on material properties low moisture content can lead to the development of electrostatic charges and influences the flow [10].”

  1. Line 51: references.

Response:

The citation has been given as required by the instruction for authors.

  1. Line 54: is the equation correct?

Response:

As we have found the referenced encyclopedia had the presented the named equation in incorrect form, we have replaced the equation with original form of Brown-Richards equation and we have changed the reference with the original paper by Brown and Richards (Lines 61-65 in the revised manuscript)

  1. Line 63: ‘Spherical and smooth particles have sufficient flow.’, it is not accurate. If particle size is very small, flow can be an issue.

Response:

Statemant in line 63 original manuscript:

Spherical and smooth particles have sufficient flow”

Has been replaced by (Line 72 in the revised manuscript)

“Spherical and smooth particles with appropriate size have sufficient flow”

  1. Line 68: reference

Response:

The reference has been properly listed ( line 77 in the revised manuscript).

  1. Line 79: how is the ‘projected conic section’ created? If the materials and methods are presented in the later section, here it should be mentioned.

Response:

Line 63 original manuscript:

“Flow curve is based on capturing real-time image data of projected conic section , which changes rapidly due to particles’ movement.”

Has been replaced by (Lines 91-92 in the revised manuscript)

“Flow curve is based on capturing real-time image data of projected conic section (described below in section 4.3.), which changes rapidly due to particles’ movement.”

  1. Line 83: ‘Measured flow curve is proportional to the changing height of the cone.’ This statement is correct only if the powder is free-flowing or easy-flowing.

Response:

Line 83 original manuscript:

Measured flow curve is proportional to the changing height of the cone, so finding the relationship between the growing height and the volume of the cone shaped particles pile helps to understand the theoretical background of the non-linear fitting.”

Has been replaced by: (Lines 95-98 of the revised manuscript)

“In a case of free or easy flowing powders measured flow curve is proportional to the changing height of the cone, so finding the relationship between the growing height and the volume of the cone shaped particles pile helps to understand the theoretical background of the non-linear fitting.”

  1. Line 85: It is questionable on the correlation between the powder flow properties and the ‘non-linear fitting’ model proposed here.

Response:

In this study this is what we have examined: Is the non-linear model correct for the powder flow characterization? And we have found that the parameters of non-linear fitting can be very useful in the characterization of powder flow properties.

  1. Line 89: 'y' is the measured area, what is the ‘measured area’?

 Response:

Line 89 in the original manuscript has been modified to:

“Where 'y' is the measured area (defined in section 4.3.), 't' is the time, 'a’ and ‘b’ are constants, ‘c’ is scaling exponent.” (line 102-103 in the revised manuscript)

  1. Line 133-136: the statement is not accurate, which depends on many factors.

Lines 133-136 in the original manuscript:

“Same weight of particles with good flowability produce lower piles than particles with poor flowability, conversely particles with good flowability produce higher piles than particles with poor flowability within a definite interval of time.”

Has been replaced by: (lines 146-148 in the revised manuscript)

“Same weight of examined particles with good flowability produce lower piles than particles with poor flowability while within a definite interval of time particles with good flowability produce higher piles than particles with poor flowability.”

  1. Data in Table 1: In principle, free-flowing materials will have a low angle of repose and a short flow time (discharge time from a funnel). In the table, the data sounds not following the rules especially for the 13-15 samples.

Response:

The test samples were assembled according to an experimental design in which four parameters significantly influencing the flow properties of the particles were varied. During our flowability  tests, we also observed that these parameters, which were varied from sample to sample according to the experimental design, could in some cases cause the flow time and the static angle of repose not to correlate closely, which we explain by the fact that these two different metrics are not always changing synchronously by changing i.e. the moisture content, particle size and added glidants.

  1. Line 203: Sphericity of the particles have been mentioned many times in the manuscript. However, there is not measurement data presented in the manuscript.

Response:

The sphericity of the examined particles has been presented in the lines 346-347 of the revised manuscript.

  1. Table 3: in the data, is the comma used for a dot as decimals?

Response:

Comma has been replace for dots.

  1. Line 240: it should be ‘the’ area under the flow curve.

Response:

The line 240 in original manuscript has been corrected as suggested, it is in the line 253 in revised manuscript.

  1. Line 249: what is the sphericity of the granule and the pellets?

Response:

The definition of sphericity of the granule and the pellets has been added in section 4.3:

(lines 431-440 in the revised manuscript.

“4.3. Sphericity determination

Sphericity (Ψ) of granules and pellets was determined by microscopic examination at 160x magnification (Carl Zeiss Axio Imager A1 Microscope, Germany) using a 5 megapixel camera (Carl Zeiss AxioCam MRc 5, Germany), examining n=50 particles both from granules:and pellet. Digital image analysis was carried out by Carl Zeiss Axio Vision Rel. 4.7 software by the following formula:

Where Af is the 2D projected area of the randomly selected particles, PCr is their Crofton perimeter. According to the sphericity calculation, a perfect round shape particle’s spericity index is 1.”

  1. Line 314-317: the physical properties of the ingredients should give, also including the granule and the pellets.

Response:

The lines 314-317 in the original manuscript has been modified as suggested

The modified paragraph is in the lines 331-341

“Pellets and granules of different sizes were produced containing 10% ibuprofen (Hungaropharma, Budapest, Hungary, average particle size 70 μm, residual moisture content 4.2%), 37% α-lactose-monohydrate (DC, BDI, Zwolle, The Netherlands, average particle size 70 μm, residual moisture content 5.1%), 50% microcrystalline cellulose (Avicel PH 101, FMC, Philadelphia, USA, average particle size 50 μm, residual moisture content 3.49%), and 3% ethylcellulose (Hercules, Wilmington, USA, 7 cP viscosity grade with 48.0–49.5 w/w of ethoxyl group). Purified water was used as granulation liquid. Quality of all materials used in the experiments was Ph. Eur.

The categorical factor for pellets and granules was distinguished on the basis of sphericity. Particles with a sphericity above 0.91 were considered pellets and those with a sphericity equal or below 0.90 were considered granules.”

  1. Line 402: what are the ‘other less important factors’?

Response:

The line 402 in the original manuscript:

“However, the cone-growth model could also determine the effect of other less important factors.”

has been modified with listing of factors with lower significance. (lines 450-452 in the revised manuscript:

“However, the cone-growth model could also determine the effect of other less important factors like electrostatic charge, chemical nature of material, temperature and powder density.”

  1. Comments on the Quality of English Language

English needs to improve accordingly. The formatting of the paper is not adequate.

Response:

English Language has been checked and edited by Grammarly.

 The structure of the paper do not follow the template file chapters, as we have found that separating chapter 2. Results and Discussion in two separate chapters would make our paper hard-to-follow, this we ask for the acceptance of current structure.

Round 2

Reviewer 2 Report

Comments and Suggestions for Authors

Authors have addressed all the comments so article is ready for publication.

Comments on the Quality of English Language

English grammar is fine. 

Author Response

Dear Reviwer!

Thank you for your valuble time, suggestions and comments. 

Reviewer 3 Report

Comments and Suggestions for Authors

This paper has been improved accordingly. However, there are a number of further comments and suggestions that need to be aware of as follows.

1, Any currents made in the revision should be marked or highlighted by red font. It is hard to identify the corrections made.

2, Figure 1: Be aware of the size of the figure.

3, Equations 3-8 can be simplified by selecting the important equations. Angle a needs to be included in a bracket otherwise it is not clear.

Comments on the Quality of English Language

English needs to double check. 

Author Response

Dear Reviwer!

Thank you for your time and precious suggestions; please find our responses to further comments in the attached file.
